# Magic in twisted transition metal dichalcogenide bilayers

Trithep Devakul [1,2✉], Valentin Crépel[1], Yang Zhang [1,2] & Liang Fu [1✉]

The long-wavelength moiré superlattices in twisted 2D structures have emerged as a highly tunable platform for strongly correlated electron physics. We study the moiré bands in twisted transition metal dichalcogenide homobilayers, focusing on $WSe_2$, at small twist angles using a combination of first principles density functional theory, continuum modeling, and Hartree-Fock approximation. We reveal the rich physics at small twist angles $\theta < 4°$, and identify a particular magic angle at which the top valence moiré band achieves almost perfect flatness. In the vicinity of this magic angle, we predict the realization of a generalized Kane-Mele model with a topological flat band, interaction-driven Haldane insulator, and Mott insulators at the filling of one hole per moiré unit cell. The combination of flat dispersion and uniformity of Berry curvature near the magic angle holds promise for realizing fractional quantum anomalous Hall effect at fractional filling. We also identify twist angles favorable for quantum spin Hall insulators and interaction-induced quantum anomalous Hall insulators at other integer fillings.

[1] Department of Physics, Massachusetts Institute of Technology, Cambridge, MA 02139, USA. [2] These authors contributed equally: Trithep Devakul, Yang Zhang. ✉email: tdevakul@mit.edu; liangfu@mit.edu

In condensed matter physics, simple and elegant models have often brought new ideas and started new paradigms. Celebrated examples include the Hubbard model for strongly correlated electron system[1], the Tomonaga-Luttinger model for one-dimensional electron liquid[2,3], and the Kitaev model for non-Abelian quantum spin liquid[4], to name a few. As toy models are designed to illustrate key concepts in the simplest form, they are rarely realized directly in real materials, whose atomic-scale electronic structures are inevitably more complex. The recent advent of long-wavelength moiré superlattices based on 2D van der Waals structures provides a new and promising venue for physical realization and quantum simulation of model Hamiltonians. In magic-angle twisted bilayer graphene[5] (TBG), experiments have discovered a variety of correlated electron states[6–12] facilitated by flat moiré bands.

More recently, moiré superlattices of semiconducting transition metal dichalcogenides (TMD) have attracted interest as a potentially simpler and more robust platform for simulating the Hubbard model on an emergent lattice[13–32]. Each lattice site represents a low-energy electronic orbital in the moiré unit cell that spreads over many atoms. These semiconductor moiré systems can thus be viewed as artificial 2D solids—a periodic array of "magnified atoms"[19]. The atomic potential depth and interatomic bonding are highly tunable by the choice of TMD materials, the twist angle[13] and the displacement field[20,23]. Thus, TMD-based moiré materials provide a favorable platform for simulating idealized models in two dimensions.

In this work, we predict the realization of generalized Kane−Mele models with topological flat band, interaction-driven Haldane insulator and Mott insulators in twisted TMD homobilayers at small twist angles. Contrary to current thoughts, we show by band structure calculation and analytical derivation that a magic twist angle exists in twisted TMD homobilayers, where the topmost valence miniband from the $\pm K$-valleys is almost perfectly flat and well separated from other bands. This band carries a spin/valley Chern number and is well described by a generalized Kane−Mele model[33].

At half filling of this topological flat band, we show that repulsive interactions drives spontaneous spin/valley polarization leading to Haldane's quantum anomalous Hall insulator[34]. We further find an out-of-plane displacement field drives a transition from the Haldane insulator into a Mott insulator. Depending on the twist angle, this Mott state is either a spin/valley polarized ferromagnet or features intervalley coherence that spontaneously breaks the spin/valley $U(1)$ symmetry. Thus our work reveals a rich phase diagram of topological, correlated and broken-symmetry insulators enabled by the flat band in TMD homobilayers at small twist angles below the 4°–5° range in current experimental studies[23,35].

Due to spin-valley locking[36], monolayer TMDs such as WSe$_2$ and MoTe$_2$ feature top valence bands with spin-↑ at $+K$ valley and spin-↓ at $-K$. We study TMD homobilayers with a small twist angle $\theta$ starting from AA stacking, where every metal (M) or chalcogen (X) atom on the top layer is aligned with the same type of atom on the bottom layer. In such twisted structure, the $K$ points of the two layers are slightly displaced and form the two corners of the moiré Brillouin zone, denoted as $\kappa_\pm$. A set of spin-↑ (↓) moiré bands is formed from hybridized $+K$ ($-K$) valley bands of the two layers. The complete filling of a single moiré band including spin degeneracy thus requires 2 holes per moiré unit cell.

## Results

In order to obtain accurate moiré band structures, we perform large-scale density functional theory calculations with the SCAN+rVV10 van der Waals density functional[37], which captures the intermediate-range vdW interaction through its semilocal exchange term. Focusing on twisted bilayer WSe$_2$, we find that lattice relaxation has a dramatic effect on moiré bands. Our DFT calculations at $\theta = 5.08°$ with 762 atoms per unit cell show a significant variation of the layer distance $d$ in different regions on the moiré superlattice, as shown in Fig. 1b. $d = 6.7$ Å is smallest in MX and XM stacking regions, where the metal atom on top layer is aligned with chalcogen atom on the bottom layer and vice versa, while $d = 7.1$ Å is largest in MM region where metal atoms of both layers are aligned. With the fully relaxed structure, the low-energy moiré valence bands of twisted bilayer WSe$_2$ are found to come from the $\pm K$ valley (shown in Fig. 1c), as opposed to the $\Gamma$ valley in previous computational studies[38] and consistent with recent works[35,39,40].

At small twist angles, the large size of moiré unit cell makes it difficult to perform DFT calculations directly on twisted TMD homobilayers. An alternative and complementary approach, introduced by Wu et al.[14], is the continuum model based on an effective mass description, which models the formation of moiré bands using spatially modulated interlayer tunneling $\Delta_T(\mathbf{r})$ and layer-dependent potential $\Delta_{1,2}(\mathbf{r})$. The continuum model Hamiltonian for $\pm K$ valley bands is given by

$$\mathcal{H}_\uparrow = \begin{pmatrix} -\frac{\hbar^2(\mathbf{k}-\kappa_+)^2}{2m^*} + \Delta_1(\mathbf{r}) & \Delta_T(\mathbf{r}) \\ \Delta_T^\dagger(\mathbf{r}) & -\frac{\hbar^2(\mathbf{k}-\kappa_-)^2}{2m^*} + \Delta_2(\mathbf{r}) \end{pmatrix} \quad (1)$$

and $\mathcal{H}_\downarrow$ as its time-reversal conjugate.

The continuum model approach is valid at small twist angle where the moiré wavelength is large enough. In this case, the atom configuration within any local region of a twisted bilayer is identical to that of an untwisted bilayer with one layer laterally shifted relative to the other by a corresponding displacement vector $\mathbf{d}_0$. For example, $\mathbf{d}_0 = 0, -(\mathbf{a}_1 + \mathbf{a}_2)/3, (\mathbf{a}_1 + \mathbf{a}_2)/3$, with $\mathbf{a}_{1,2}$ the primitive lattice vector of a monolayer, correspond to the MM, MX and XM regions respectively. Therefore, the moiré potentials for twisted TMD bilayers $\Delta_T(\mathbf{r})$ and $\Delta_{1,2}(\mathbf{r})$ as a function of coordinate on the moiré superlattice can be determined from the valence band edges of the untwisted bilayer as a function of the corresponding shift vector[20]. In the lowest harmonic approximation, $\Delta_T(\mathbf{r})$ and $\Delta_{1,2}(\mathbf{r})$ are sinusoids that interpolate between MM, MX and XM regions[14]:

$$\Delta_{1,2}(\mathbf{r}) = 2V \sum_{j=1,3,5} \cos(\mathbf{g}_j \cdot \mathbf{r} \pm \psi) \quad (2)$$

$$\Delta_T(\mathbf{r}) = w(1 + e^{-i\mathbf{g}_2 \cdot \mathbf{r}} + e^{-i\mathbf{g}_3 \cdot \mathbf{r}}) \quad (3)$$

where $\mathbf{g}_j$ are $(j-1)\pi/3$ counter-clockwise rotations of the moiré reciprocal lattice vector $\mathbf{g}_1 = (4\pi\theta/\sqrt{3}a_0, 0)$, and $a_0$ is the monolayer lattice constant. Up to an overall energy scale, the continuum model depends only on the dimensionless parameters $\alpha \equiv V\theta^2/(m^* a_0^2)$, $w/V$ and $\psi$.

From our DFT calculation for untwisted bilayers with relaxed layer distance and using the vacuum level as an absolute reference energy for the band edge, we obtain the continuum model parameters $V = 9.0$ meV, $\psi = 128°$, and $w = 18$ meV. Importantly, the interlayer tunneling strength $w$ is twice larger than previously reported[14]. To demonstrate the accuracy of the continuum model method, we compare in Fig. 1c the band structures computed by large-scale DFT directly at $\theta = 5.08°$ and by the continuum model with the above parameters, finding excellent agreement. Details on the DFT calculation can be found in Supplementary Note 1 (see supplemental material for details on DFT calculations and continuum model).

We remark that different approaches[31,32,40] can lead to different conclusions on topology. Thus, we utilize a method to determine band topology directly from our large-scale DFT band

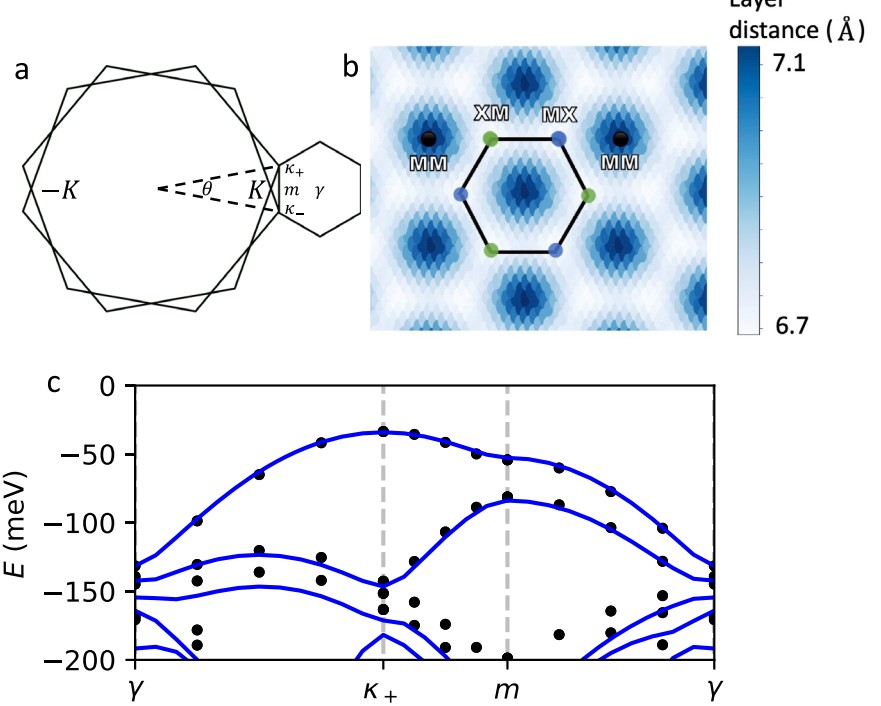

**Fig. 1 Comparison with large-scale DFT calculations. a** The $\kappa_\pm$ points of the moiré Brillouin zone are formed from the $K$ points of the monolayer Brillouin zones, which are rotated by $\pm\theta/2$. **b** The interlayer distance of the twisted WSe$_2$ structure obtained from DFT is shown, demonstrating a large variation between the MM and XM/MX regions. **c** The continuum band structure (blue lines) is plotted in comparison with large-scale DFT calculations (black dots) at twist angle $\theta = 5.08°$, showing excellent agreement.

**Table 1 $C_{3z}$ eigenvalues of the first two bands from each valley, computed from large-scale DFT wavefunctions at high symmetry momentum points.**

| Band, Valley | $\kappa_+$ | $\kappa_-$ | $\gamma$ |
|---|---|---|---|
| 1, K | $e^{i\pi/3}$ | $e^{i\pi/3}$ | $e^{i\pi}$ |
| 1, K' | $e^{-i\pi/3}$ | $e^{-i\pi/3}$ | $e^{i\pi}$ |
| 2, K | $e^{-i\pi/3}$ | $e^{-i\pi/3}$ | $e^{i\pi/3}$ |
| 2, K' | $e^{i\pi/3}$ | $e^{i\pi/3}$ | $e^{-i\pi/3}$ |

structure based on symmetry eigenvalues. As detailed in Supplementary Note 2 (see supplemental material for details of topology of DFT wavefunctions), we are able to isolate bands from the $\pm K$ valley and compute their $C_{3z}$ eigenvalue at the high symmetry momenta $\gamma$, $\kappa_\pm$, which determine their Chern number (mod 3)[41]. The $C_{3z}$ eigenvalues for the first two bands, summarized in Table 1, are consistent with the first two bands having non-trivial valley Chern number $\mathcal{C}_{K,1} = \mathcal{C}_{K,2} = 1$.

Using the new continuum model parameters established above, along with the lattice constant $a_0 = 3.317\,\text{Å}$[42] and the effective mass $m^* = 0.43 m_e$[43,44], we calculate the band structure of twisted bilayer WSe$_2$, $E_i(\mathbf{k})$, at various twist angles, as shown in Fig. 2a. The bandwidth of the first band, $W = \max_{\mathbf{k}} E_1(\mathbf{k}) - \min_{\mathbf{k}} E_1(\mathbf{k})$, as well as the (direct or indirect) band gaps $\varepsilon_{ij}$ between pairs of bands $(i,j) = (1,2)$ and $(2,3)$, $\varepsilon_{ij} = \min_{\mathbf{k}} E_i(\mathbf{k}) - \max_{\mathbf{k}} E_j(\mathbf{k})$, is shown in Fig. 3. Focusing on topological features of the first two valence bands, we can divide the moiré band structure into three main regimes divided by $\theta_1 \approx 1.5°$ and $\theta_2 \approx 3.3°$.

First, for $\theta < \theta_1$, the top two bands are well separated from the rest of the spectrum, and carry opposite Chern number $[\mathcal{C}_{K,1}, \mathcal{C}_{K,2}] = [+1, -1]$. The bandwidth of the first band $W < 1$ meV remains very small throughout. In this regime of very small

twist angles, the character of the top two valence bands can be understood from an effective tight binding model on a moiré honeycomb lattice that takes the form of a Kane−Mele model, as suggested in the insightful work of Wu et al.[14] As we will later show, the original Kane−Mele description with up to second nearest neighbor hopping terms only describes the band structure well for very small angles $\theta \lesssim 1°$. As $\theta$ increases towards $\theta_1$, longer range hopping terms become more important.

At $\theta = \theta_1$, the band gap $\varepsilon_{23}$ closes and the Chern number of the top two bands changes to $[+1, +1]$. In this second regime, $\theta_1 < \theta < \theta_2$, both top bands have same Chern number $[+1, +1]$ and are still all separated by a sizable gap $\varepsilon_{12}, \varepsilon_{23} > 0$. The bandwidth of the first band increases rapidly with $\theta$, reaching around $W \approx 20$ meV at $\theta_2$ (not shown). Finally, in the third regime, $\theta_2 < \theta \lesssim 5.4°$, the indirect gap $\varepsilon_{12}$ vanishes, but the direct gap remains open. The Chern number of the top two bands remains well defined at $[+1, +1]$, but the bands now overlap in energy and are highly dispersive. In both the second and third regimes, $\varepsilon_{23} > 0$; thus, the top two bands together form a gapped $\mathcal{C} = 2$ manifold. Beyond $\theta \gtrsim 5.4°$, the gap $\varepsilon_{23}$ also vanishes and the top two bands are no longer isolated (Supplementary Note 3; see supplemental material for derivation of the analytic magic angle). Topology of the continuum model at $\theta \approx 5°$ is consistent with that determined directly from large-scale DFT in Table 1, further strengthening our confidence in the continuum model description even up to larger angles.

For $\theta < \theta_2$, especially near $\theta_2$ where the first band is more dispersive, the spin Chern number $\mathcal{C} = 1$ and $\varepsilon_{12} > 0$ is favorable for a quantum spin Hall insulator at a filling of $n = 2$ holes per moiré unit cell. Also, for the wide range of angles $\theta_1 < \theta \lesssim 5.4°$, $\varepsilon_{23} > 0$ and the top two bands both carry spin Chern number $\mathcal{C} = 1$, giving rise to a double quantum spin Hall state with two sets of counter-propagating spin-polarized edge modes at $n = 4$.

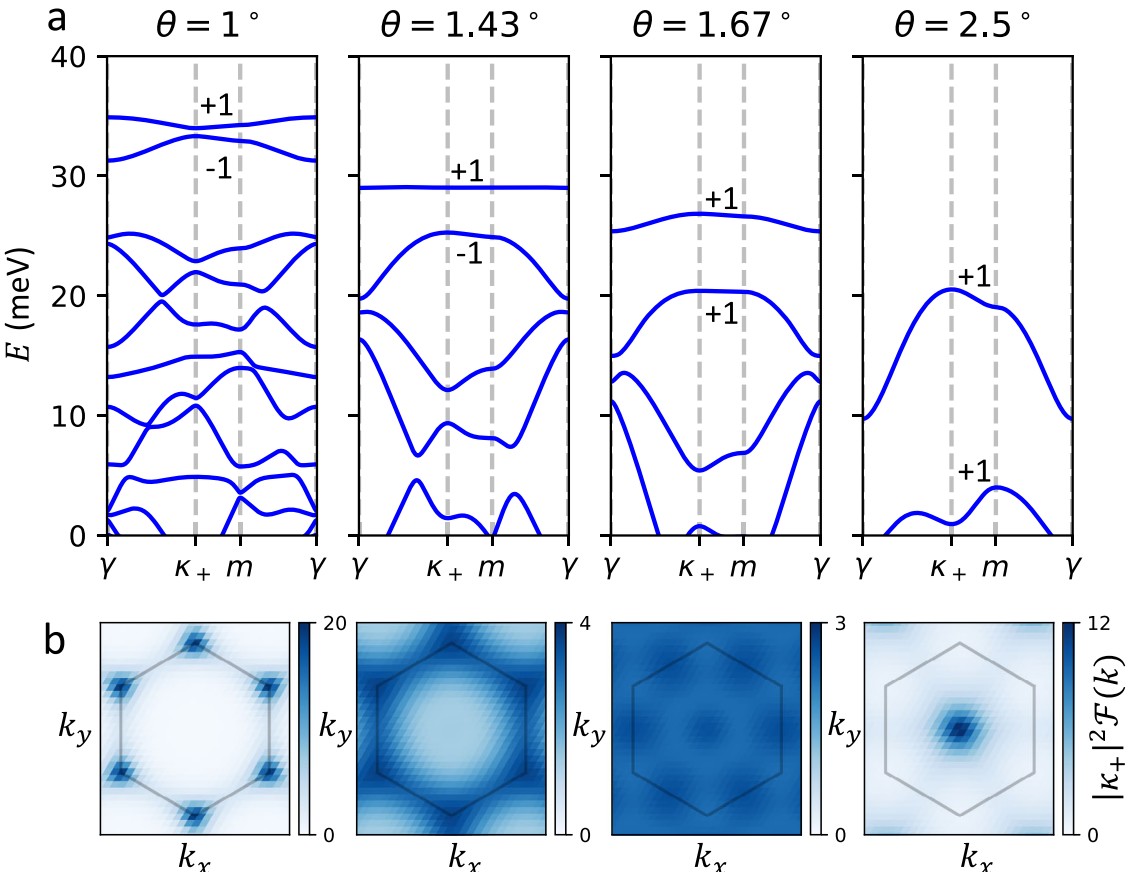

**Fig. 2 Continuum model band structure and Berry curvature at various twist angles. a** The band structure $E_i(\mathbf{k})$ along with the Chern numbers of the first two bands and (**b**) the (scaled) Berry curvature $|\kappa_+|^2\mathcal{F}(\mathbf{k})$ of the first band is shown for the continuum model at $\theta = 1°$, $1.43°$, $1.67°$, and $2.5°$. At $\theta = 2.5°$, the first band maxima is located at the $\kappa_\pm$ points and the Berry curvature is peaked at $\gamma$. At $\theta = 1°$, the band maximum is instead at $\gamma$ and $\mathcal{F}$ is peaked at $\kappa_\pm$. During the crossover region between these angles, $E_1(\mathbf{k})$ and $\mathcal{F}(\mathbf{k})$ both become very flat. We find that the band dispersion $E_1(\mathbf{k})$ is flattest at $\theta \approx 1.43°$ and the Berry curvature $\mathcal{F}(\mathbf{k})$ is most uniform at $\theta \approx 1.67°$, both shown.

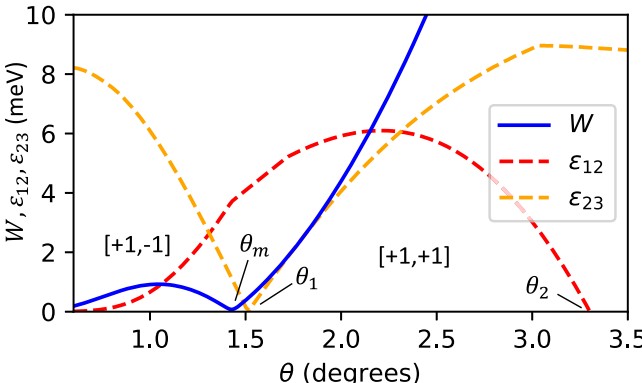

**Fig. 3 The bandwidth of the first band $W$, and indirect band gap between the first two pairs of bands $\varepsilon_{12}$ and $\varepsilon_{23}$.** The bandwidth is minimized at $\theta_m$, while being well separated from the remaining bands. The Chern numbers of the first two bands, $[\mathcal{C}_{K,1}, \mathcal{C}_{K,2}]$, is shown before and after the $\varepsilon_{23}$ gap closing at $\theta = \theta_1$. For $\theta \geq \theta_2$, $\varepsilon_{12}$ vanishes.

We now address the bandwidth $W$, which shows a sharp minimum at $\theta = \theta_m \approx 1.43°$ reminiscent of the magic angle in TBG. To understand this, notice that the top band, shown in Fig. 2a, has two qualitatively different behaviors in the small and large $\theta$ limit. For $\theta \gtrsim 2.5°$, $E_1(\mathbf{k})$ reaches its maximum at $\kappa_\pm$ and minimum at $\gamma$, which can be understood from the weak moiré

effects at small $\alpha$. For small $\theta \lesssim 1°$, the opposite holds and $E_1(\mathbf{k})$ is maximum is at $\gamma$ and minimum at $\kappa_\pm$, which can be understood from the effective Kane−Mele model, which we will derive explicitly. At the crossover between these two limits, the band maxima and minima must switch locations in the moiré Brillouin zone, potentially leading to a flat band. As can be clearly seen, $W$ achieves a minimum at a particular magic angle $\theta_m$ during this crossover. At $\theta_m$ the gap to the next state $\varepsilon_{12} \approx 3.7$ meV is much larger than the bandwidth $W \approx 0.1$ meV. The band structure at $\theta_m$ is shown in Fig. 2a, which shows that the first band is almost completely flat and separated from the next band. For even smaller $\theta$, both $\varepsilon_{12}$ and $W$ vanish, but the ratio $W/\varepsilon_{12}$ diverges. Thus, we may view $\theta_m$ as the angle at which the top band is flattest while still being well isolated from the rest of the spectrum.

Analytic progress can be made in estimating the magic angle by considering the dispersion near $\gamma$. Assuming that the bandwidth will be minimized near the angle at which $E_1(\gamma)$ changes from minima to maxima, expanding $E_1(\gamma + \mathbf{k}) \approx E_1(\gamma) + \frac{k^2}{2m_\gamma} + \mathcal{O}(k^3)$, the effective mass $m_\gamma$ should diverge near the crossover. Let $\tilde{\theta}_m$ to be the angle at which $m_\gamma^{-1} = 0$. Then, considering only the six most relevant states at $\gamma$, we have (Supplementary Note 4; see supplemental material for continuum model band structure at higher twist angles)

$$\tilde{\theta}_m^{-2} = \frac{8\pi^2}{9m^*a_0^2}\left(\frac{1}{\mathcal{E}_{n_0} - \mathcal{E}_{n_0+1}} + \frac{1}{\mathcal{E}_{n_0} - \mathcal{E}_{n_0-1}}\right) \quad (4)$$

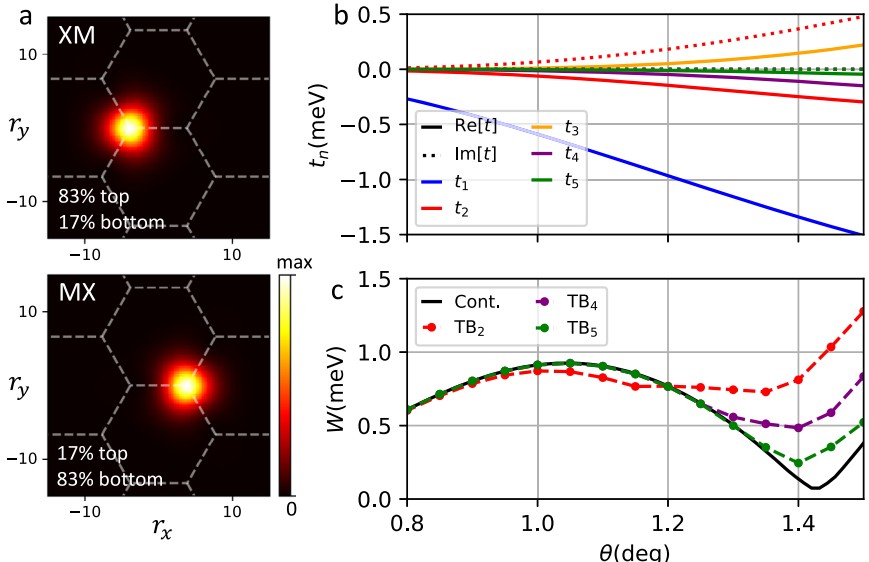

**Fig. 4 Wannier functions and tight binding model parameters. a** Wannier functions at the magic angle, **b** tight binding parameters as a function of $\theta$, and **c** the bandwidth of the top band in the effective tight binding model TB$_n$ keeping up to $n$th nearest neighbor hopping terms, compared to that of the continuum model.

where $\mathcal{E}_n = 2w\cos(\pi n/3) + 2V\cos(2\pi n/3 - \psi)$, and $n_0$ is the integer (mod 6) which maximizes $\mathcal{E}_n$ ($n_0 = 1$ for WSe$_2$ parameters). We find that Eq. (4) provides a decent estimate for $\theta_m$ in the cases considered. In WSe$_2$, we have $\tilde{\theta}_m = 1.47°$, compared to $\theta_m = 1.43°$ at which the bandwidth is minimized.

Next, we turn to the Berry curvature $\mathcal{F}(\mathbf{k})$ of the top band, shown in Fig. 2b. In all cases, the first band has Chern number $\mathcal{C}_{K,1} = \frac{1}{2\pi}\int_{BZ}\mathcal{F}d\mathbf{k} = 1$; however, the distribution changes very drastically as $\theta$ is varied. At $\theta \gtrsim 2°$, $\mathcal{F}$ is peaked around the band minimum at $\boldsymbol{\gamma}$. At $\theta \lesssim 1°$, $\mathcal{F}$ is instead sharply peaked at the $\boldsymbol{\kappa}_{\pm}$ points. Near the crossover region, the distribution of $\mathcal{F}$ shifts from $\boldsymbol{\gamma}$ to $\boldsymbol{\kappa}_{\pm}$, and can become very evenly distributed. We find that $\mathcal{F}$ is most evenly distributed near $\theta = 1.67°$, shown in Fig. 2b, where $\mathcal{F}$ becomes almost uniform in the Brillouin zone. The uniform distribution of $\mathcal{F}$ is reminiscent to that of Landau levels. Time-reversal symmetry forces the corresponding spin-↓ bands from the $-K$ valley to have opposite Chern number.

We emphasize that the physics of the magic angle arises due to the crossover between two qualitatively different behaviors of the first band at low and high angles. Additional factors unaccounted for by the continuum model may result in, for example, angle-dependent model parameters. However, as long as the qualitative behaviors at small and large angles are unchanged, there will still be crossover regime at which the band becomes flat. Even when the bands are not perfectly flat, a diverging mass can still give rise to a diverging higher-order van Hove singularity[23].

Recall that for $\theta < \theta_1$, the top two bands carry opposite Chern number and are separated from the rest of the spectrum, suggesting a description in terms of an effective tight binding model. We now focus on $\theta < \theta_1$ and directly derive an effective tight binding model for the first two moiré bands by explicitly constructing a basis of localized Wannier states. These Wannier states are constructed via a simple procedure which manifestly preserves the symmetries of the twisted homobilayer. Given the single-particle eigenstates $\{|\phi_{n,\mathbf{k}}\rangle\}$ of the continuum Hamiltonian (1) for each $\mathbf{k}$ in the mBZ, we first construct a superposition of the first two ($n = 1, 2$) eigenstates, $|\tilde{\phi}_{n,\mathbf{k}}\rangle = \sum_{m=1,2} U_{nm}^{(\mathbf{k})}|\phi_{m\mathbf{k}}\rangle$ using a $2 \times 2$ unitary matrix $U_{nm}^{(\mathbf{k})}$, which maximizes the layer polarization at every $\mathbf{k}$:

$$P_{\mathbf{k}} = \sum_{n=1,2}(-1)^n\langle\tilde{\phi}_{n,\mathbf{k}}|(\mathcal{P}_- - \mathcal{P}_+)|\tilde{\phi}_{n,\mathbf{k}}\rangle, \quad (5)$$

where $\mathcal{P}_{\pm}$ is the projector on to the top/bottom layer, so that $|\tilde{\phi}_{1,\mathbf{k}}\rangle$ is chosen to mostly consist of states in the top layer, and similarly for $|\tilde{\phi}_{2,\mathbf{k}}\rangle$ on the bottom layer. This uniquely specifies $|\tilde{\phi}_{n,\mathbf{k}}\rangle$ up to a phase, which we choose to be real and positive at the XM ($n = 1$) or MX ($n = 2$) stacking regions (Supplementary Note 5; see supplemental material for details on the derivation of localized Wannier functions and tight binding model). The Wannier states at moiré lattice vector $\mathbf{R}$ is then defined $\left|W_{\mathbf{R}}^n\right\rangle = \frac{1}{\sqrt{N_k}}\sum_{\mathbf{k}}e^{-i\mathbf{k}\cdot\mathbf{R}}|\tilde{\phi}_{n\mathbf{k}}\rangle$. They are localized about their centers with a root-mean-square distance $a_W \approx 5$ nm, and are also mostly composed of states in one layer: $\langle W_{\mathbf{R}}^1|\mathcal{P}_+|W_{\mathbf{R}}^1\rangle \approx 0.83$ is mostly in the top layer, and vice versa for $\left|W_{\mathbf{R}}^2\right\rangle$.

It is straightforward to obtain the hopping matrix elements of the effective tight binding model in the Wannier basis for the top two bands as a function of $\theta$. Figure 4b shows the $n$th nearest neighbor hopping matrix elements $t_n$ obtained in this way, up to $n = 5$. As anticipated, the effective tight binding model at $\theta < \theta_1$, including the spin/valley degrees of freedom, is a generalized Kane−Mele model with sites centered on the honeycomb lattice formed by MX and XM stacking regions[14].

The tight binding Hamiltonian is found to be

$$\mathcal{H}_{TB} = t_1\sum_{\langle i,j\rangle,\sigma}c_{i\sigma}^{\dagger}c_{j\sigma} + |t_2|\sum_{\langle\langle i,j\rangle\rangle,\sigma}e^{i\phi\sigma\nu_{ij}}c_{i\sigma}^{\dagger}c_{j\sigma} + \cdots \quad (6)$$

where $c_{i\sigma}^{\dagger}, c_{i\sigma}$ are fermionic creation/annihilation operators, $\sigma = \pm$ is the spin/valley degree of freedom, the sum $\langle i, j\rangle$ ($\langle\langle i, j\rangle\rangle$) is over (next) nearest neighboring sites $i, j$ of the honeycomb lattice, and $\nu_{ij} = \pm 1$ depending on whether the path $i \to j$ turns right ($+$) or left ($-$). The parameter $t_1$ is real, while $t_2 \equiv |t_2|e^{i\phi}$ is complex, and $\cdots$ contain longer range hopping terms. We find that $|t_n|$ quickly reduce in magnitude with hopping distance $n$, and only the second neighbor hopping has a significant imaginary component. In Fig. 4c, we show the bandwidth of the top band, $W$, in the effective tight binding model TB$_n$ including up to $t_n$ hopping

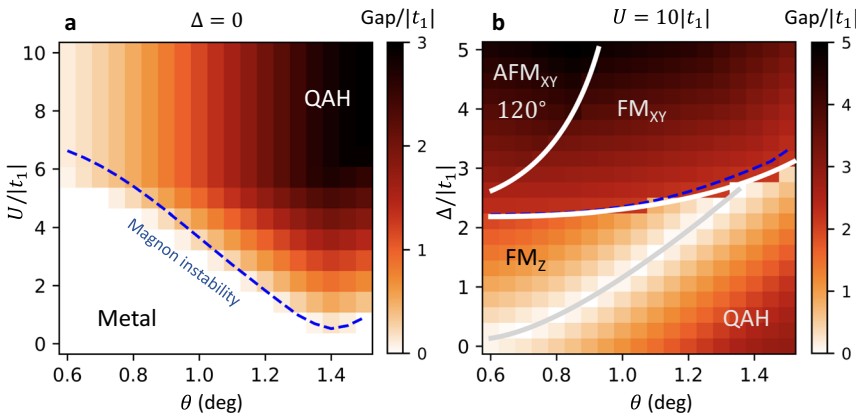

**Fig. 5 Numerical solution of the self-consistent Hartree−Fock approximation.** The phase diagram is shown as a function of the twist angle and (**a**) interaction strength at fixed displacement field $\Delta = 0$, or **b** displacement field at fixed interaction strength $U = 10|t_1|$, including up to fifth nearest neighbor hopping terms. The insulating phases denoted by QAH, $FM_z$, 120° $AFM_{xy}$, and $FM_{xy}$ are described in the main text. The dashed blue line shows the boundary of the $FM_z$ phases determined by the first magnon instability. The colors indicate the charge gap. Hartree−Fock calculations were done with a $\sqrt{3} \times \sqrt{3}$ unit cell of 6 atom sites, and a $30 \times 30$ grid of ($\mathbf{k}$) points.

terms, compared to that of the continuum model. For $\theta \lesssim 1°$, $TB_2$ already captures the band structure very well. Near the magic angle, higher range hoppings become more important in capturing the flatness of the band.

For the small twist angles $\theta < \theta_1$ considered, since the size of the Wannier orbitals are small compared to the moiré unit cell, the dominant interaction is a simple on-site Hubbard term $\mathcal{H}_U = U\sum_i n_{i\uparrow} n_{i\downarrow}$. We also estimate $U \sim e^2/(\epsilon a_W) \approx 70$ meV at $\theta_m$, using a realistic relative dielectric constant $\epsilon = 4$ and $a_W = 5$nm, which is significantly larger than the tight binding parameters $t_n$. Therefore at such small twist angle, twisted $WSe_2$ homobilayers are in the strong-coupling regime, in contrast with $\theta \sim 4°–5°$ where the bandwidth is comparable to the interaction strength[23,35].

In the following, we shall focus on the strongly correlated regime $\theta < \theta_1$ at a filling of $n = 1$ holes per moiré unit cell, where we expect the flat bands will favor the quantum anomalous Hall (QAH) insulator due to spontaneous spin/valley polarization. The reason is as follows. First, spin/valley polarized states filling the top band of $\mathcal{H}_{TB}$ with $\sigma = \pm$ are exact eigenstates of our interacting model, because the spin-orbit coupling in $\mathcal{H}_{TB}$ conserves the $z$-spin/valley component. Then, these spin/valley polarized states avoid the Hubbard interaction, and also minimize the kinetic energy in the case of a completely flat top band. Minimizing both parts of the Hamiltonian, they necessarily are many body ground states of the model at $n = 1$ filling. The complete polarization of $\mathcal{H}_{TB}$ exactly corresponds to Haldane's model for QAH insulator.

To include corrections coming from the finite bandwidth $W$ of the flat band, we solve our interacting problem within the Hartree−Fock (HF) approximation, where the Hubbard interaction is decoupled as $n_{i\uparrow} n_{i\downarrow} \simeq \langle n_{i\uparrow} \rangle n_{i\downarrow} + \langle n_{i\downarrow} \rangle n_{i\uparrow} - \langle c_{i\uparrow}^\dagger c_{i\downarrow} \rangle c_{i\downarrow}^\dagger c_{i\uparrow} - \langle c_{i\downarrow}^\dagger c_{i\uparrow} \rangle c_{i\uparrow}^\dagger c_{i\downarrow}$, up to a constant term, and where the expectation values for the spin and density at each site are determined self-consistently by iteration. Our numerical solutions of the HF equations are shown in Fig. 5a as a function of twist angle and interaction strength. As expected, we observe a transition from a metallic state to a ferromagnetic QAH insulator polarized along $z$ when $U$ increases. Within HF, this transition can be understood as follows. The fully polarized states yield a rigid shift of the bands by $\sigma U/2$. When $U$ is larger than the non-interacting bandwidth $W$, a full gap opens and the ferromagnetic state fully fills one of Chern bands of $\mathcal{H}_{TB}$, which leads to a QAH phase. We remark that the appearance of the QAH phase relies on

both the non-trivial Chern number as well as the fact that the band is flat and isolated, features which are maximized at the magic angle, as illustrated by a dip of the insulating phase above $\theta \simeq 1.4°$.

To precisely locate the transition between QAH insulator and the metallic phase, we compute the magnon excitation spectrum above the fully ferromagnetic state by exact diagonalization (ED) of the interacting Hamiltonian projected on the spin-1 excitation subspace[45]. For large $U$, this spectrum is gapped and the QAH is robust against spin flips. Decreasing $U$ eventually brings one magnon at zero energy, which destabilizes the ferromagnetic states and drives the transition to a metal. As shown in Fig. 5a, the ED results almost perfectly agrees with the HF boundaries, putting them on firmer grounds.

For the large values of $U$ relevant to $WSe_2$, the magnons have a large gap, and the lowest excitation corresponds to an interband transition between two bands with same spin. The QAH phase is thus protected by a gap $\varepsilon_{12} \approx 3.7$ meV near the magic angle, leading to quantum Hall effect at elevated temperature.

We also highlight that the QAH may also be observed for larger twist angles, where the first band still carries a non-zero Chern number (Fig. 2), and its bandwidth remain small compared to the estimated $U$ (Fig. 3). Likewise, the second band is topological and quite flat for $\theta \sim 2°–3°$ (Supplementary Note 3), and therefore QAH may also be observed at a filling of $n = 3$. Twisted TMD bilayers with $\pm K$-valley bands are thus expected to be an intrinsically robust platform for interaction-induced QAH phases.

It is interesting and worthwhile to compare the QAH phase in twisted TMD and graphene bilayers. Anomalous Hall effect and its quantization have been experimentally observed in magic-angle TBG[11,12], where the alignment with hBN substrate is likely the origin of valley Chern number[46] and both spin and valley degeneracy are lifted due to repulsive interaction in the flat band[47–51]. Due to the presence of $SU(2)$-invariant spin degrees of freedom, QAH in TBG is subject to the adverse effect of gapless thermal fluctuation, which forbids long-range order at finite temperature in the thermodynamic limit. In contrast, spin-valley locking in TMD systems enables robust Ising-type spin/valley order that leads to QAH effect at lower temperature.

Another great advantage of twisted TMD bilayer is their high degree of tunability, in particular with respect to applied electric fields[19,35,52]. Due to the layer polarization of the Wannier basis states, the displacement field can be modeled as a sublattice

symmetry breaking term $\mathcal{H}_\Delta = \frac{\Delta}{2}\sum_i s_i c_{i\sigma}^\dagger c_{i\sigma}$, where $s_i$ is $(-)1$ for $i$ in the $A$ $(B)$ sublattice. Including this term in our HF treatment, we can investigate which phases should neighbor the QAH ferromagnet in experiments. We present our solutions of the HF equations as a function of twist angle and displacement field in Fig. 5b. There, we fix $U = 10|t_1|$, a tradeoff between the large $U$ of the homobilayer system and the convergence rate of the HF self-consistent iteration algorithm. We find it necessary to consider an enlarged $\sqrt{3}\times\sqrt{3}$ unit cell in order to describe all ordered phases of the model.

At small displacement fields, the topmost moiré band remain relatively flat and our earlier arguments for spin/valley polarization still apply. This is confirmed by our HF solutions for $\Delta \lesssim 2t_1$ (Fig. 5b), which exhibit full spin polarization along the $z$ axis. In this region, a transition nevertheless occurs at $\Delta = 6\sqrt{3}|t_2|\sin\phi$ (up to $t_{n\geq3}$ terms), where the single-particle gap between the two moiré bands closes, and their Chern numbers change from $[+1, -1]$ to $[0, 0]$. This gap closing line marks the transition between a QAH insulator and a topologically trivial ferromagnet with spin/valley polarization (FM$_z$)[34]. As the displacement field further increases, the bandwidth $W$ also grows, which decreases the magnon gap (see discussion above). The spin/valley polarized phases eventually become unstable when the magnon gap closes, which can be seen with the very good agreement between the phase boundaries determined with HF and ED (Fig. 5b).

Beyond this spin-wave instability line, our results show the emergence of two new Mott insulating phases, where holes are mostly localized on the $A$ sublattice, and their spin either form an antiferromagnetic pattern (120° AFM$_{xy}$), or ferromagnetically align in the $xy$ plane (FM$_{xy}$). Their appearance is most easily understood for large displacement fields, where the physics becomes analogous to that of localized moments on the triangular $A$ sublattice. In the regime $t_2 \lesssim t_1 \ll \Delta$, $U$ relevant for our system, their coupling is described by an effective XXZ model with Dzyaloshinskii−Moriya (DM) interactions

$$\mathcal{H}_S = \sum_{\langle i,j\rangle_B} J_\parallel s_i^z s_j^z + J_\perp(s_i^x s_j^x + s_i^y s_j^y) + D\left[(\mathbf{s}_i\times\mathbf{s}_j)\cdot\mathbf{z}\right], \quad (7)$$

which is derived in Supplementary Note 6 (see supplemental material for the derivation of the effective spin Hamiltonian). The parameters of this effective spin model are given by

$$J_\parallel = \frac{4|\tilde{t}|^2}{U} + \mathrm{Re}\left(\frac{4t_1^2 t_2}{\Delta^2}\right), \quad (8a)$$

$$J_\perp = \mathrm{Re}\left(\frac{4\tilde{t}^2}{U} + \frac{4t_1^2 t_2}{\Delta^2}\right), \quad (8b)$$

$$D = \mathrm{Im}\left(\frac{4\tilde{t}^2}{U} + \frac{4t_1^2 t_2}{\Delta^2}\right), \quad (8c)$$

with $\tilde{t} = t_2 + t_1^2/\Delta$. In Eq. 8, we have separated exchange terms coming from different physical processes. The first ones $\propto \tilde{t}^2/U$ arise from nearest neighbor tunneling on the triangular $A$ sublattice, while the others $\propto t_2(t_1/\Delta)^2$ originate from loop-exchange on the honeycomb lattice that do not involve any double occupancy.

For twist angles $\theta \lesssim 1°$, $t_2 \ll t_1$ and Eq. (7) reduces to an antiferromagnetic (AFM) Heisenberg model, where $J_\parallel = J_\perp > 0$ are dominated by the nearest neighbor tunneling on the triangular lattice. This simplified triangular lattice description, valid for very small twist angles, has been proposed in earlier studies of Mott insulators in twisted TMDs[24,26,53]. It was shown to yield an antiferromagnetic phase that the small residual DM interaction

pins in the $xy$ plane. This is the origin of the AFM$_{xy}$ phase observed in Fig. 5b. We also note that the weak-coupling version of AFM$_{xy}$ phase—an intervalley-coherent $\sqrt{3}\times\sqrt{3}$ density wave[23]—has been proposed for the correlated insulating state at $n = 1$ in twisted bilayer WSe$_2$ at $\theta \sim 4°-5°$[35].

For larger twist angles, $t_2$ becomes substantial and we observe that $J_\perp$ becomes negative for the realistic parameter $U \gg \Delta$, dominated by a third-order exchange process $\propto t_1^2 t_2$ on the honeycomb lattice without double occupancy. Then, the FM$_{xy}$ phase is favored as shown in Fig. 5b. The competition between AFM$_{xy}$ and FM$_{xy}$ phases can be analyzed by solving Eq. (7) for classical spins. This approach, detailed in the Supplementary Note 6 (see supplemental material for the derivation of the effective spin Hamiltonian), gives a transition between the two phases when $|D| = -\sqrt{3}J_\perp$. For $\Delta = 5t_1$, this criterion yields a critical twist angle $\theta = 0.95°$, which roughly agrees with our HF results. We note that the ferromagnetic phase due to $J_\perp < 0$ does not appear in twisted TMDs based on simplified triangular lattice descriptions.

Finally, we comment on the effect of nearest neighbor repulsion $V\sum_{\langle i,j\rangle}n_i n_j$ to our HF phase diagram. This term favors the layer polarized phases, such as the FM$_{xy}$ and AFM$_{xy}$ which appear at large $|\Delta|$. Small $V$ therefore narrows the range in $\Delta$ at which the QAH phase appears. For large $V$, there is a sharp transition at $\Delta = 0$ between layer polarized Mott insulating phases, which can lead to the strong hysteretic behavior of Mott ferroelectricity[20,54]. The long-range component of interactions can be controlled by screening from nearby metallic layers. Multiple recent experiments[55,56] on twisted WSe$_2$ homobilayers in the presence of a nearby WSe$_2$ monolayer report strong screening effects when the monolayer is doped. This raises the interesting possibility of a screening-induced transition between the QAH and Mott insulating phases.

## Discussion

Our phase diagram demonstrates the high experimental tunability of TMD twisted homobilayers, where the applied displacement field can tune between quantum anomalous Hall phase and Mott insulators involving three types of magnetic orders: FM$_z$, FM$_{xy}$, and AFM$_{xy}$. Despite being electrically insulating, the $xy$-ordered Mott insulators support coherent magnon transport[23], which can be detected by optical spin injection and spatial-temporal mapping recently developed for TMD bilayers[57]. The experimental feasibility of tuning and distinguishing between topologically different insulators at the same filling adds to the attractiveness and desirability of TMD-based moiré systems.

In parallel to our work on twisted TMD homobilayers, a breakthrough experiment led by Kin Fai Mak and Jie Shan discovered unexpectedly a QAH phase with spontaneous spin/valley polarization in a TMD heterobilayer MoTe$_2$/WSe$_2$ at the filling of $n = 1$ tuned by displacement field[58,59]. Large-scale DFT calculation and wavefunction analysis reveal two dispersive moiré bands forming the Kane−Mele model, suggestive of a similar origin of QAH as described here.

Looking forward, the remarkable flat Chern band we found, combined with the uniformity of Berry curvature, suggests that twisted TMD homobilayers near magic angle may be an ideal setting for observing a fractional quantum anomalous Hall state at zero magnetic field.

## Data availability

The data needed to evaluate the conclusions in the paper are present in the paper and the Supplementary Material. The full dataset generated during this study, including relaxed lattice structure and band structure obtained from DFT, tight binding model parameters, and self-consistent HF solutions, has been deposited in the Zenodo database[60]. Additional

data related to this paper are available from the corresponding authors upon reasonable request.

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

## Acknowledgements

We thank Kin Fai Mak, Jie Shan, Tingxin Li, and Shengwei Jiang for ongoing collaborations on MoTe$_2$/WSe$_2$, Bi Zhen and Constantin Schrade for previous collaborations on related topics. We thank Pablo Jarillo-Herrero, Kenji Yasuda, Cory Dean, Abhay Pasupathy, Qianhui Shi, Augusto Ghiotto, and En-Min Shih for helpful discussions. This work is primarily supported by the DOE Office of Basic Energy Sciences, Division of Materials Sciences and Engineering under Award DE-SC0020149 (band structure calculation), DE-SC0018945 (theoretical modeling), and Simons Investigator award from the Simons Foundation (numerical analysis). L.F. is partly supported by the David and Lucile Packard Foundation.

## Author contributions

T.D., V.C., Y.Z. and L.F. performed research, analyzed data, and wrote the manuscript.

## Competing interests

The authors declare no competing interests.
