## [Peer Review File · Nature Communications]

Magic in twisted transition metal dichalcogenide bilayersREVIEWER COMMENTS

Reviewer #1 (Remarks to the Author):

The manuscript by T. Devakul et al. reports a computational and theoretical study of electronic structure in twisted bilayer WSe₂. In my opinion, most of the results described in this manuscript has already been known in the literature, in particular, (1) twisted TMD homobilayer may realize Kane-Mele model and (2) Kane-Mele-Hubbard model could support quantum anomalous Hall effect at the filling factor $\nu=1$. It is fair to say that the results obtained in this manuscript are only incremental. Moreover, the approximations used in treating the large-scale moire superlattices are open to question. Therefore, I am not convinced that this manuscript should be published in Nature Communications regarding novelty as well as validity. My comments on the technical part is in the following.

(1) Can the authors verify that the DFT band structure shown in Fig. 1(c) is topological?

(2) It seems that different approaches can lead to different conclusions about the topological nature of the moire bands in twisted bilayer WSe₂. See, for examples, arXiv:2101.04867, arXiv:2103.07447, and arXiv:2106.06058. In arXiv:2106.06058, it is shown that the K-valley valence band edge behaves differently for twist angle below and above 1 degree. In the former case, the band maximum appears at MX and XM regions of the superlattice, forming a honeycomb lattice and implying topological bands. In the latter case, interlayer hybridization dominates, shifting the band maxima to MM regions, forming a triangular lattice and implying topologically trivial bands. Can the authors make comments on the different approaches and conclusions?

(3) The parameters in the continuum model are obtained by fitting to the DFT band structure. Does the fitting lead to a unique determination of the parameter values? It seems not so as shown in arXiv:2103.07447.

(4) Even if the fitting accurately reproduces both the band energy and wave function of the DFT bands in Fig. 1(c), can the same set of parameter values be applied to other twist angles? As pointed out in arXiv:2106.06058 and many other works, there can be significant lattice relaxation effects in twisted bilayer WSe₂ for small twist angles.

(5) For the interacting Hamiltonian, can off-site repulsion be neglected? This approximation seems not justified, given the observation of abundant electronic crystal states in TMD moire systems. In particular, the nearest-neighbor repulsion U_1 may stabilize a sublattice polarized Mott insulating state at $\nu=1$, which spontaneously breaks the honeycomb lattice symmetry. as proposed in arXiv:2009.14224 by some of the same authors. This state can compete with the quantum anomalous Hall insulator and could be more favorable due to the tiny bandwidth compared to the sizable U_1 . This casts doubt about the relevance of the phase diagram shown in Fig. 5a.

Reviewer #2 (Remarks to the Author):

In this work the authors study the electronic properties of twisted bilayers of WSe₂. They start with a DFT calculation of a unit cell with ~ 800 atoms including vW correction. Next the energy bands are fitted to a $k \cdot p$ model derived by Wu et al. in Ref.10. The parameters of the $k \cdot p$ model are used to obtain energy bands for holes at small twisted angles with unit cells so large as not being accessible to ab-initio calculations. The authors fully analyzed the energy bands as a function of twist angle and show the emergence of flat bands for holes. In particular, existence of a magic angle for which the width of the band reaches minimum. They use the two bands to construct Wannier functions and derive the tight binding Hamiltonian for a moire superlattice. The authors show that the effective TB

Hamiltonian is that of a Kane-Mele model. They next use the HF approximation and magnon spectrum to show the existence of Haldane and Mott insulators and quantum anomalous Hall effect and other phases of interacting electrons at different filling of the bands.

The methodology appears correct and results appear valid. Overall this is a well written paper on a timely and interesting topic and I recommend publications after some minor corrections listed below.

Some minor corrections include:

1) references to Hubbard, Tomonaga-Luttinger and Kitaev would help the reader entering this field.

2) Fig 2 needs axis labels

3) Fig 3 – here energy bands are at positive energies but in Fig. 2 and $k \cdot p$ Hamiltonian they are negative?

4) on page 4 we read:

“It is straightforward to obtain the hopping matrix elements of the effective tight binding model in the Wannier basis for the top two bands as a function of μ . Figure 4b shows the n th nearest neighbor hopping matrix elements t_n obtained in this way”

Something is missing here ...

5) on page 4 we read:

“parameter t_1 is real, while $t_2 = jt_2jei$ is complex, and 288 $_{-} _{-} _{-}$ contain longer range hopping terms”

Again, Something is missing here ...

----- Reply to Reviewer #1-----

Comment:

Reviewer Comments:

Reviewer #1 (Remarks to the Author):

The manuscript by T. Devakul et al. reports a computational and theoretical study of electronic structure in twisted bilayer WSe₂. In my opinion, most of the results described in this manuscript has already been known in the literature, in particular, (1) twisted TMD homobilayer may realize Kane-Mele model and (2) Kane-Mele-Hubbard model could support quantum anomalous Hall effect at the filling factor $\nu=1$. It is fair to say that the results obtained in this manuscript are only incremental. Moreover, the approximations used in treating the large-scale moire superlattices are open to question. Therefore, I am not convinced that this manuscript should be published in Nature Communications regarding novelty as well as validity. My comments on the technical part is in the following.

We thank the reviewer for their report and assessment of our manuscript. However, we respectfully disagree with their determination that our manuscript is only incremental. The reviewer neglects to mention of our main result – the identification of a magic angle condition in moiré TMD bilayers. This is contrary to current belief in the literature and opens many avenues for future studies of twisted TMDs. The analytic form for the magic angle we have derived means that our result can also be extended generically to other systems and will be valuable in guiding future experiments.

Regarding the reviewer's claim that some of our results are already known: (1) we are not aware of any other work which derives the generalized Kane-Mele model parameters directly from the continuum model. Wu et al (Ref [10] in the original manuscript) only suggest the Kane-Mele model as an effective description, but the parameters are not derived directly from Wannier functions as we have done. In addition, the model we derived also contains a significant real second neighbor hopping as well as many further range hopping, which is absent in the original Kane-Mele models previously studied. Thus, we do not simply show that the Kane-Mele-Hubbard model supports QAH as the reviewer suggests in (2). Rather, we show that a honeycomb lattice tight binding description of our system at small twist angles, which takes the form of a highly generalized Kane-Mele-Hubbard model, realizes QAH along with many other Mott insulating phases. The fact that we find many phases (FM_z, FM_{xy}, AFM_{xy}, QAH) all existing in the phase diagram easily tunable by displacement field in the vicinity of the magic angle, is another important result directly relevant to experiments.

In response to an insightful question by the reviewer, we have determined the non-trivial topology of the moiré bands directly from large-scale DFT. Our method, detailed in the supplemental material, can be applied to future studies and further adds to the novelty of our manuscript.

Regarding the validity of our methods, we refer to the answers for the specific questions below.

(1) Can the authors verify that the DFT band structure shown in Fig. 1(c) is topological?

We have now computed the C_{3z} eigenvalue of the first two bands of each valley directly from large-scale DFT at the high symmetry points $\gamma, \kappa_+, \kappa_-$, using the method which we have added to the supplemental material. They are summarized in the revised manuscript and in the table below:

Band, Valley	C_{3z} at κ_+	C_{3z} at κ_-	C_{3z} at γ
1, K	$e^{\frac{\pi}{3}i}$	$e^{\frac{\pi}{3}i}$	$e^{\pi i}$
1, K'	$e^{-\frac{\pi}{3}i}$	$e^{-\frac{\pi}{3}i}$	$e^{\pi i}$
2, K	$e^{-\frac{\pi}{3}i}$	$e^{-\frac{\pi}{3}i}$	$e^{\frac{\pi}{3}i}$
2, K'	$e^{\frac{\pi}{3}i}$	$e^{\frac{\pi}{3}i}$	$e^{-\frac{\pi}{3}i}$

From these eigenvalues, it follows that these bands are all topological. A non-topological band, which can be described by a localized Wannier orbital must necessarily have either: the same C_{3z} eigenvalue at all three momenta (corresponding to a Wannier center at the MM region, chosen as the origin), or all three different C_{3z} eigenvalues $e^{\pi i}, e^{\pm \frac{\pi}{3}i}$ (corresponding to a Wannier center at the MX or XM region). We see that this is neither the case for any of these bands. The relative C_{3z} eigenvalues at these three momenta determine the Chern number mod 3, and is consistent with the continuum model prediction that both bands have valley-Chern number $C_{K,1} = C_{K,2} = 1$. This further supports the validity of our continuum model description of the DFT band structure.

We have added a discussion of this to the manuscript.

(2) *It seems that different approaches can lead to different conclusions about the topological nature of the moire bands in twisted bilayer WSe₂. See, for examples, arXiv:2101.04867, arXiv:2103.07447, and arXiv:2106.06058. In arXiv:2106.06058, it is shown that the K-valley valence band edge behaves differently for twist angle below and above 1 degree. In the former case, the band maximum appears at MX and XM regions of the superlattice, forming a honeycomb lattice and implying topological bands. In the latter case, interlayer hybridization dominates, shifting the band maxima to MM regions, forming a triangular lattice and implying topologically trivial bands. Can the authors make comments on the different approaches and conclusions?*

Let us address each of these references individually:

In [arXiv:2101.04867], the authors find, using a modified continuum approach for relaxed WSe₂, qualitatively similar results as ours in the angle range $0.8^\circ < \theta < 2.1^\circ$, including the topology of the first two bands. At very small or large angles $\theta < 0.8^\circ$, $\theta > 2.1^\circ$, they report a topologically trivial first band. While the energy bands are validated against large-scale DFT at 5.08° , the topology of the DFT bands are not. In our answer to (1), we directly showed the non-trivial topology of our fully relaxed DFT band structure at 5.08° . This lends further credence to our continuum model parameters, which correctly captures the DFT band topology even at the large angle 5.08° .

In [arXiv:2103.07447], the authors perform large-scale DFT in the atomic orbital basis using lattice structure relaxed from classical force fields, which allows them to go to smaller angles. We are unclear on their method of calculating Chern numbers from their DFT. In contrast, our large-scale DFT is performed in the plane wave basis which is less restrictive, and the structure is obtained from ab initio lattice relaxation. Thus, our results should be more accurate. For $\theta > 4.4^\circ$, they find the first two bands carry opposite valley Chern number [+1, -1], which disagrees with our DFT results [see answer to (1)].

In [arXiv:2106.06058], the authors show that for twist angles above 1° , that the band edges are centered at the MM regions and topologically trivial. This is done through an effective model containing many different effects (“interlayer hybridization, lattice reconstruction, piezoeffects, and interlayer charge transfer”), with parameters extracted from (untwisted) DFT calculations with shifts.

Our main concern with this work is that the many different effects may lead to overfitting of parameters, which the band topology depends sensitively on. They also do not confirm their results with any large-scale DFT. Their conclusion is in disagreement with our fully relaxed large-scale DFT calculation at 5.08° (as well as [arxiv: 2103.07447]), which clearly shows MX and XM localized wavefunctions.

All these works are cited in the revised manuscript.

(3) The parameters in the continuum model are obtained by fitting to the DFT band structure. Does the fitting lead to a unique determination of the parameter values? It seems not so as shown in arXiv:2103.07447.

We do *not* obtain the parameters to the continuum model by fitting to the large-scale DFT band. We use the shift method which, as mentioned in the main text (and elaborated on in the supplemental material), obtains the continuum model parameters from DFT calculations of commensurate structures at high symmetry shifts MM, MX, and XM. This unambiguously determines the continuum model parameters. Fig. 1c in the main text illustrates a comparison between the continuum model parameters obtained from the shift method and large-scale DFT at 5.08° : two independent methods whose agreement serves to establish confidence in our continuum model. Related to our answer to question (1), the fact that the topology agrees at 5.08° further boosts our confidence in the continuum model.

(4) Even if the fitting accurately reproduces both the band energy and wave function of the DFT bands in Fig. 1(c), can the same set of parameter values be applied to other twist angles? As pointed out in arXiv:2106.06058 and many other works, there can be significant lattice relaxation effects in twisted bilayer WSe₂ for small twist angles.

Our continuum model parameters were obtained from the fully relaxed structure at high symmetry stackings. Thus, our continuum model parameters accurately describe the structure with fully relaxed interlayer distance at small angles. Our large-scale DFT calculation at 5.08° is also fully relaxed. The fact that the energy bands and topology of the continuum model both agree with fully relaxed large-scale DFT at 5.08° is remarkable, and supports the notion that our continuum model correctly captures the correct band structure and topology in this whole range of angles.

Additional relaxation effects (such as those considered in [arXiv:2101.04867]) may result in angle-dependent continuum model parameters. However, this would not change the main result of our paper. While this may affect the value of the magic angle, the essential physics of the magic angle will remain. As we mention in the manuscript, the flat band at magic angle arises from the crossover between qualitatively different band structure at small and large angles. We expect honeycomb lattice-like bands at small angles (as suggested by Wu et al [10], and in agreement with arXiv:2106.06058 below 1°) at which γ is a band maximum of the first band. At large twist angles, our continuum model (and large-scale DFT) observes a band minimum at γ , as is expected at larger angles from the nearly-free hole dispersion at K, K'. In between these two limits, there must be angle at which the effective mass at γ diverges, at which the first band should be flat. In the case of angle dependent parameters, our analytic expression (Eq 4) should still hold, except both sides of the equation are θ dependent.

We have added some discussion to the manuscript regarding this.

(5) For the interacting Hamiltonian, can off-site repulsion be neglected? This approximation seems not justified, given the observation of abundant electronic crystal states in TMD moire systems. In particular, the nearest-neighbor repulsion U_1 may stabilize a sublattice polarized Mott insulating state at $\nu=1$, which spontaneously breaks the honeycomb lattice symmetry, as proposed in arXiv:2009.14224 by some of the same authors. This state can compete with the quantum anomalous

Hall insulator and could be more favorable due to the tiny bandwidth compared to the sizable $U1$. This casts doubt about the relevance of the phase diagram shown in Fig. 5a.

We thank the reviewer for raising this excellent point. Next nearest neighbor interactions will indeed favor a layer polarized state and compete with the QAH phase. Below, we show the valley and layer polarization of the tight binding model at $\theta = 1.4^\circ$ computed using infinite density matrix renormalization group (IDMRG) for an infinite cylinder of circumference $L=6$ unit cells, well converged at bond dimension $\chi = 800$, in the presence of a nearest neighbor repulsion V . We observe a first order phase transition at $V = 1.5\text{meV}$ from the fully valley polarized QAH phase to the FM_{xy} phase, which is mostly layer (sublattice) polarized.

The nearest neighbor (and longer range) interactions can be efficiently screened by the presence of nearby metallic layers. These can be metallic gates or even a nearby third TMD monolayer as in recent experiments [arxiv:2108.06588] and [arxiv:2108.07131], in which the screening strength is tuned by the doping of the monolayer. In [arxiv: 2108.06588], a third WSe_2 is placed 2-4 hBN layers away (corresponding to roughly 1-2 nm). Taking a simple screened Coulomb interaction $V_s(r) = \frac{e^2}{\epsilon r} e^{-\frac{r}{\xi}}$ with $\epsilon = 4\epsilon_0$, a simple estimate for the nearest neighbor repulsion is $V = V_s\left(\frac{a_M}{\sqrt{3}}\right)$. Below, we show V as a function of the screening length ξ . As can be seen, a screening layer $\sim 1-2$ nm away should be sufficient to screen interaction enough for the QAH phase to appear. This also raises an interesting experiment in which QAH can be tuned via screening. When screening is weak, spontaneous sublattice polarization results in the interesting physics of Mott ferroelectricity.

We have added a discussion of this to the manuscript.

Reviewer #2 (Remarks to the Author):

In this work the authors study the electronic properties of twisted bilayers of WSe₂. They start with a DFT calculation of a unit cell with ~ 800 atoms including vW correction. Next the energy bands are fitted to a $k \cdot p$ model derived by Wu et al. in Ref.10. The parameters of the $k \cdot p$ model are used to obtain energy bands for holes at small twisted angles with unit cells so large as not being accessible to ab-initio calculations. The authors fully analyzed the energy bands as a function of twist angle and show the emergence of flat bands for holes. In particular, existence of a magic angle for which the width of the band reaches minimum. They use the two bands to construct Wannier functions and derive the tight binding Hamiltonian for a moire superlattice. The authors show that the effective TB Hamiltonian is that of a Kane-Mele model. They next use the HF approximation and magnon spectrum to show the existence of Haldane and Mott insulators and quantum anomalous Hall effect and

other phases of interacting electrons at different filling of the bands.

The methodology appears correct and results appear valid. Overall this is a well written paper on a timely and interesting topic and I recommend publications after some minor corrections listed below.

We thank the reviewer for the thorough reading and clear summary of our work. We appreciate that the reviewer understands the importance of our work and thus recommends publication in Nature Communications. We respond to the specific comments below.

Some minor corrections include:

1) references to Hubbard, Tomonaga-Luttinger and Kitaev would help the reader entering this field.

Following this suggestion, we have added these references.

2) Fig 2 needs axis labels

We have added axis labels to this figure.

3) Fig 3 - here energy bands are at positive energies but in Fig. 2 and $k \cdot p$ Hamiltonian they are negative?

We have changed the axis label in Fig 3 in order to make it clear that we are plotting the energy gaps, which are defined to be positive.

4) on page 4 we read:

“It is straightforward to obtain the hopping matrix elements of the effective tight binding model in the Wannier basis for the top two bands as a function of θ . Figure 4b shows the n th nearest neighbor hopping matrix elements t_n obtained in this way”

Something is missing here ...

5) on page 4 we read:

“parameter t_1 is real, while $t_2 = |t_2| e^{i\theta}$ is complex, and t_{288} contain longer range hopping terms”

Again, Something is missing here ...

We could not find anything missing in these sentences. Perhaps there is a technical issue with the pdf file.

The first should read:

“It is straightforward to obtain the hopping matrix elements of the effective tight binding model in the Wannier basis for the top two bands as a function of θ .”

And the second should read
“... is complex, and ... contain longer range hopping terms”,
where the dots “...” refers to the terms left out in Equation 6.

REVIEWERS' COMMENTS

Reviewer #2 (Remarks to the Author):

The authors addressed my comments. I also examined critique by the first referee and authors response. The twisted bilayer field attracts a lot of attention. The present work identifies magic angle physics and potentially contributes to our understanding of these systems. I maintain my recommendation to publish.

REVIEWERS' COMMENTS

Reviewer #2 (Remarks to the Author):

The authors addressed my comments. I also examined critique by the first referee and authors response. The twisted bilayer field attracts a lot of attention. The present work identifies magic angle physics and potentially contributes to our understanding of these systems. I maintain my recommendation to publish.

We thank Reviewer 2 for the positive feedback.